# A New Toolbox in Experimental Embryology—Alternative Model Organisms for Studying Preimplantation Development

**DOI:** 10.3390/jdb9020015

**Published:** 2021-04-02

**Authors:** Claudia Springer, Eckhard Wolf, Kilian Simmet

**Affiliations:** 1Chair for Molecular Animal Breeding and Biotechnology, Gene Center and Department of Veterinary Sciences, Ludwig-Maximilians-Universität München, 85764 Oberschleissheim, Germany; c.springer@gen.vetmed.uni-muenchen.de (C.S.); ewolf@genzentrum.lmu.de (E.W.); 2Laboratory for Functional Genome Analysis (LAFUGA), Gene Center, Ludwig-Maximilians-Universität München, 81377 Munich, Germany; 3Center for Innovative Medical Models (CiMM), Ludwig-Maximilians-Universität München, 85764 Oberschleissheim, Germany

**Keywords:** embryo, cattle, pig, ART, SCNT, genome editing

## Abstract

Preimplantation development is well conserved across mammalian species, but major differences in developmental kinetics, regulation of early lineage differentiation and implantation require studies in different model organisms, especially to better understand human development. Large domestic species, such as cattle and pig, resemble human development in many different aspects, i.e., the timing of zygotic genome activation, mechanisms of early lineage differentiations and the period until blastocyst formation. In this article, we give an overview of different assisted reproductive technologies, which are well established in cattle and pig and make them easily accessible to study early embryonic development. We outline the available technologies to create genetically modified models and to modulate lineage differentiation as well as recent methodological developments in genome sequencing and imaging, which form an immense toolbox for research. Finally, we compare the most recent findings in regulation of the first lineage differentiations across species and show how alternative models enhance our understanding of preimplantation development.

## 1. Introduction 

To study the events during preimplantation development, a look beyond the most commonly used mouse model can be vital to discover the often still unknown molecular pathways that regulate the first steps of embryo development. Where the mouse shows unique regulatory mechanisms, other animals share great similarities in their developmental plan. The moment of zygotic genome activation, the first lineage differentiations and maintenance of pluripotency are some aspects that are not always conserved between species, but are very similar in cows and pigs compared to humans (reviewed in [1,2,3]). Furthermore, the scarcity of human embryos and the ethical and logistical challenges increase the need to work with other models. At first glance, working with large domestic species to study preimplantation development may appear laborious and impractical. However, looking closer, there are numerous benefits that come from highly developed assisted reproductive technologies (ARTs) in these species. To produce embryos regularly, it is sufficient to have a nearby abattoir, where ovaries can be obtained. Using in vitro techniques, an unlimited amount of research material can be produced without the need of housing experimental animals. If embryos at developmental stages beyond our current in vitro culture capabilities are required, protocols for embryo transfer (ET) and recovery are available. 

With recent breakthroughs in genome editing, it is now possible to perform a wide range of reverse genetics studies in large domestic species in a time and resource efficient manner. In combination with the different ARTs that are available, a plethora of possible studies may be conducted to increase our knowledge about mechanisms and dynamics during preimplantation development in alternative model organisms. Here, we describe in detail the different ARTs and their benefits or disadvantages for studying preimplantation development and we show, when and how manipulations of the embryo can be performed to shed light on the regulation of the first lineage differentiations.

## 2. State of the ART—Assisted Reproductive Technologies in Cattle and Pigs

Artificial insemination (AI) and other ARTs have revolutionized the cattle and pig industry. The use of AI enabled tremendous genetic improvement of dairy cows by dissemination of superior bulls, increasing the milk yield per year 3.8-fold from 2400 kg to 9200 kg in only 57 years (1950–2007) in the USA [4]. The pig industry increased the number of piglets weaned per sow per year from 20 to 30 over the last three decades [5]. ARTs, i.e., ET, ovum pick-up (OPU) and in vitro fertilization (IVF), have also been developed to increase the number of offspring of uniparous animals such as cows ([6,7], reviewed in [8]). These techniques enable the use of cattle and pigs as model organisms for developmental biology and biomedical research. In this chapter, we concentrate on ARTs in cows and pigs and provide an overview about the different techniques, the advantages and limitations of these procedures.

### 2.1. Superstimulation and Embryo Transfer

The principle of superstimulation regarding its commercial use is to increase the number of offspring of cows with superior breeding values. Multiple oocytes ovulate and after AI, the uterus of the superstimulated cow is flushed and the obtained embryos are transferred to recipients. Hence, more calves with superior genetics will be born in a shorter time range. For this purpose, follicle-stimulating hormone (FSH) or similar gonadotropins are administered. After ovulation, superstimulated females release large numbers of oocytes into the oviduct which are then fertilized via AI, develop in vivo and can be flushed non-surgically directly from the uterus (reviewed in [9]). Additionally, it is possible to collect in vivo matured oocytes by flushing the oviduct, but a surgical [10,11,12] or transvaginal endoscopic approach [13] is necessary. Although the ovarian response varies a lot among individual donors and treatment protocol, this technique enormously increases the numbers of retrieved embryos or oocytes. Two extensive studies showed an average of 6.9 embryos collected from beef cows [14] and 5.1 to 5.4 viable embryos from lactating dairy cows [15]. According to the Association of Embryo Technology in Europe (AETE), an average of 6.9 embryos per collection from dairy and beef cows was achieved in 2019 [16]. An indicator for the population of antral follicles in both human and cow is anti-Müllerian hormone (AMH). Concentrations of AMH in the plasma may predict a cow’s response to superovulatory treatment [17,18]. 

The fertilization rate after AI in heifers is decreased after superstimulation compared to spontaneous ovulation (72% vs. >80%), suggesting an impact on oviductal function [19]. Effects on embryos derived from superovulation procedures were investigated by Gad, et al. [20], illustrating a reduced competence for preimplantation development in vivo and altered gene expression patterns.

In pigs, superovulation is only rarely performed, as it is a multiparous species. In breeds with physiologically low ovulation rates, such as Duroc, it can help to increase the embryo yield, with normal embryonic and fetal development [21,22].

With ET, it is possible to remove one or more embryos from the reproductive tract of a donor female and subsequently transfer them to surrogates. The nonsurgical transfer into the bovine uterus is the standard technique when using either fresh or cryopreserved day 7 blastocysts. Of importance is the synchronous reproductive cycle of the recipients (reviewed in [8]). If earlier stages need to be transferred, Besenfelder and Brem [23] developed a transvaginal endoscopic technique to insert early tubal stage embryos (day 1–2) into the oviduct. This technique allows embryos to passage through the oviduct during the period when major epigenetic reprogramming and major embryonic genome activation take place ([24,25], reviewed in [26]). These processes are easily disturbed by changes in the environmental conditions [27]. Interestingly, transfer of early cleavage stage embryos into the uterus of domestic animals leads to impaired development and results in low pregnancy outcomes [28], whereas in humans, uterus transfer with zygotes or early cleavage stage embryos is commonly performed [29].

In pigs, ET is often used in combination with somatic cell nuclear transfer (SCNT) to produce genetically modified livestock, using a minimal invasive laparoscopic method ([30], reviewed in [31]). Additionally, a nonsurgical method for deep uterine embryo transfer was established, which could pave the way for a commercial use [32,33].

### 2.2. Ovum Pick-Up (OPU)

In 1988, a Dutch team first collected cumulus-oocyte complexes (COCs) from cattle by using transvaginal ultrasound-guided follicle aspiration, giving rise to a new procedure called OPU [7]. OPU is combined with in vitro production (IVP) of bovine embryos and is an alternative to superstimulation. There are many advantages: in contrast to superovulation, the reproductive status of the donor is irrelevant, it can even be pregnant, acyclic, or having genital tract infections. Furthermore, heifers that are not responding to the superstimulation treatment can be used as well. As OPU can be performed twice a week, it can increase the yield of transferable embryos immensely [34]. It is performed regularly over a long period and donors with a high number of COCs seem to perform steadily on a high level [35]. Still, between breeds and different animals, the number of retrieved COCs per OPU session is variable [36]. The collected COCs are then used for IVP of embryos (see Section 2.5).

### 2.3. Intrafollicular Oocyte Transfer (IFOT)

Recently, a technique for intrafollicular oocyte transfer (IFOT) in cows has been established [37]. Here, immature COCs derived from abattoir ovaries or by OPU are transferred directly into a pre-ovulatory follicle of synchronized heifers to enable maturation in vivo prior to AI. This procedure circumvents the disadvantages of in vitro maturation (IVM) of oocytes and results in higher blastocyst rates (40.1 vs. 29.3% after IFOT and IVM, respectively) [38]. IFOT allows the production of a high number of embryos in a complete in vivo system without any hormonal superstimulation or extensive laboratory facilities [37,39]. However, pregnancy rates were rather low when using cryopreserved embryos derived from IFOT (15.4%) [38].

### 2.4. Intracytoplasmic Sperm Injection (ICSI)

Intracytoplasmic sperm injection (ICSI) describes a microfertilization technique of the direct injection of a single spermatozoon or sperm head (nucleus) into the ooplasm. It is possible to use immobilized or dead sperm, making it especially interesting as an alternative to in vitro fertilization to overcome male infertility in humans (reviewed in [40]). The first offspring from ICSI-derived embryos was described by Martin [41] in pigs and by Goto, et al. [42] in cattle. In bovine, ICSI has not been established for commercial use, as IVF protocols are very efficient; therefore, it is used for research interest only [36]. The same is true for ICSI in pigs, where costs and effort cannot be compensated by the low success rates, which makes it impractical for pig production [41,43,44,45,46]. Nevertheless, as polyspermy is a common phenomenon in IVF in pigs (see Section 2.5), ICSI is a considerable alternative (reviewed in [40]). Furthermore, ICSI-mediated gene transfer can be used for genetic modification of porcine embryos (see Section 3). Besides humans and pigs, ICSI is merely interesting for horses, because methods for capacitating sperm in vitro have not been developed so far [47].

### 2.5. In Vitro Production (IVP) of Embryos

Since many decades, IVP protocols exist for bovine embryos and they have been improved constantly, while in pig the procedure still requires improvement. The aim is to generate embryos in the laboratory via fertilization of oocytes, which have been matured either in vitro (well established for cows and pigs) or in vivo (mostly for mouse and human). In domestic species, COCs can be derived from abattoir ovaries, making it possible to procure great amounts of oocytes without much effort. If ex vivo derived oocytes are desired, superstimulation or OPU can be performed.

After collecting the COCs, the first step is their IVM. Gonadotropins, such as FSH and luteinizing hormone (LH) are supplemented to simulate the preovulatory surge of those hormones to achieve an expansion of cumulus cells and resumption of meiosis. The hormones are combined with serum, bovine serum albumin (BSA) or epidermal growth factor (EGF), which help to stimulate maturation and cumulus expansion. As serum may vary dependent on its batch, serum-free media are preferred [48]. Additionally, serum–containing media may induce a shift towards a higher proportion of male bovine embryos [49]. Subsequently, after 22–24 h (bovine) or 44–48 h (pig), the matured oocytes can be fertilized. A defined sperm concentration without undesirable semen components enables continuity in IVF procedures. In preparation of sperm for IVF, centrifugation through a Percoll density gradient is the most conventional method in cattle, but other procedures such as swim-up, centrifugation on BSA, or Sephadex column separation are available [50,51]. Heparin, which is found in the genital tract of females, supports fertilization of matured oocytes by inducing sperm capacitation [52]. Subsequently, presumptive zygotes are placed into embryo culture medium after removing excess sperm cells and cumulus cells to avoid the presence of degenerating cells that may decrease the efficiency of the culture system [50]. Seven days after insemination, bovine blastocysts can be cryopreserved or used for ET (reviewed in [8]).

For basic research, it is possible to maintain bovine embryos until day 8 or 9 in culture, when they have developed to blastocysts that hatch from the zona pellucida. Routinely achieved day 8 blastocyst rates in bovine are approximately 30–40% [12,53]. Approaches to prolong development in vitro have been elaborated recently. In the post-hatching development (PHD) system, embryos are cultured in an agarose-coated dish in serum- and glucose-enriched medium (PHD medium) until day 15 or 16, when they show epiblast (EPI)-derived cells, a Rauber’s layer and some degree of proliferation of hypoblast (HYPO) cells. Although trophectoderm (TE) cells can grow in the PHD medium, HYPO migration along the entire inner embryo surface was not achieved, apoptosis and necrosis were visible and EPI formation was compromised in this system. Therefore, PHD medium supports proliferation of the TE but is incapable to maintain embryo development beyond the blastocyst stage [54,55,56,57]. In a different approach, embryos were cultured in N2B27 medium (used in mouse and human primed and naïve stem cell culture) and reduced oxygen (5%) until day 15. Embryos were routinely obtained and showed HYPO formation and varying amounts of EPI, with several embryos displaying a SOX2 positive EPI disc [58]. Recently, a three-dimensional (3D)-printed oviduct-on-a-chip platform was created, which mimicked the oviductal environment in vitro. In this culture chamber, oviductal epithelial cells were incubated, thus fertilization and early embryo development resembled the physiological situation more closely, leading to bovine zygotes with a similar transcriptome profile compared to in vivo produced zygotes [27].

Of importance is the difference between both human and mouse compared to domestic animals regarding the peri-implantation development. In cattle, the embryo will elongate up to 20 cm via rapid trophoblast development that dramatically alters the blastocyst morphology prior to implantation and similar growth is seen in pigs, whereas in human and mouse, there is no elongation ([59,60,61], reviewed in [62]). Therefore, to study peri-implantation development in humans, large domestic animals may not serve as optimal model organisms.

In small ruminants (goats and sheep), IVP protocols are also accessible, where embryos can be cultured until day 8 with similar outcomes as in cattle ([63], reviewed in [64]).

It is important to compare IVP embryos with their in vivo equivalents. Whereas IVP shows a fertilization rate of up to 80% and a blastocyst rate of 30–40% [12,53], a fertilization rate by AI of over 90% in ovulated oocytes is described, with most of the resulting zygotes developing to blastocysts [65]. More differences comparing in vitro versus in vivo embryos are seen regarding the ultrastructure [66], microvilli [67], lipid content [68], cryoresistance [67], and most importantly the gene expression profile. Altered transcript levels in IVP embryos are connected to metabolism and growth as well as altered fetal development after transfer [69,70,71].

The large variety of media used in IVP is a problem when comparing results of different research groups. Using serum in medium can modulate the gene expression pattern and decrease cryoresistance of bovine IVP blastocysts [68,72]. Regarding bovine blastocyst yield and quality, there was no difference between media containing estrous cow serum or BSA [73].

In pig, IVP is not as developed as in cattle, leading to highly variable success rates that are below those achieved in bovine IVP [74,75,76]. Blastocysts derived by IVP procedures show an inferior number of cells and lower ability to produce pregnancies compared to their ex vivo counterparts [77]. Nevertheless, it is feasible to culture porcine embryos until day 6–8 and progress has been made in implementing 3D culture systems to investigate elongated stages [78,79,80]. A yet unresolved problem in pig IVP protocols is the high proportion of polyspermy. Imbalanced nuclear and cytoplasmatic maturation as well as a low quality of oocytes and increased sperm concentrations are discussed as factors causing polyspermic penetration of porcine oocytes. Polyspermic embryos are aneuploid, show abnormal cleavage patterns, reduced growth of the inner cell mass (ICM), and cannot develop to term, thereby decreasing the IVP efficiency [81,82,83,84].

### 2.6. Somatic Cell Nuclear Transfer (SCNT)

During SCNT, the nucleus of a somatic donor cell is introduced into an oocyte whose own nuclear DNA has been removed (enucleation). This reconstructed embryo is activated to progress embryonic development and emerging embryos can be transferred to a recipient, enabling development to term. The nuclear genome of the resulting offspring is identical to the respective donor cell, whereas the mitochondrial DNA is mostly or completely derived from the recipient oocyte [85]. In agriculture, cloning can help to preserve genetic resources and to expand the distribution of breeding livestock, reviewed in [86,87]. As genome editing efficiency has improved immensely in recent years, it is now feasible to use SCNT for producing genetically engineered (GE) livestock to enhance demanded traits such as improved product quality, rapid growth or resistance of diseases [36,88,89]. Tsunoda, et al. [90] reported a general blastocyst rate of 10–40% in bovine SCNT experiments, of which 10–30% developed into calves upon transfer to recipients. SCNT may serve as an important key tool for studying preimplantation development, when combined with gene editing procedures (see Section 3).

In pig, blastocyst rates of SCNT embryos vary between 20 and 40% [91,92,93,94], but the overall cloning efficiency—defined as the number of cloned piglets born per transferred SCNT embryos—is low at 1–5%, as shown in an extensive study over three years [95].

Despite numerous promising advantages, SCNT is not only impeded by its low efficiency, but cloned animals may also suffer from various developmental defects. Problems occurring when conducting SCNT are micromanipulation trauma, oocyte incompetence, in vitro culture-induced anomalies and failed epigenetic reprogramming of the transferred nucleus (reviewed in [96]). As a result, physiological development is considered to be impaired as abnormal epigenetic profiles and gene expression may occur (reviewed in [97]). After transfer of bovine SCNT embryos to recipients, placental failure has frequently been observed, likely due to abnormal embryo-maternal communication during peri-implantation [98,99], giving a possible explanation for the high rate of pregnancy failures. The so-called “large offspring syndrome” is connected to cloned cattle and sheep neonates with unusually large bodies and sometimes associated organ defects, but the syndrome is also described in IVP embryos [100]. In pigs, aberrant cleft lips or teat numbers were found in surviving SCNT animals [101]. Cao, et al. [102] described a delayed zygotic genome activation (ZGA) and altered gene expression patterns in pig embryos produced by SCNT. Despite its limitations, SCNT has tremendous advantages, particularly for the generation of genetically engineered/genome edited large animal models, and further progress in modulating the epigenome could improve nuclear reprogramming (reviewed in [97]).

## 3. Genetic Manipulations

A great variety of possible experiments emerges when researchers combine different ART procedures with new tools (Figure 1) which precisely edit the genome, such as CRISPR/Cas9 (reviewed in [103,104,105,106]). This RNA-guided nuclease induces double strand breaks (DSBs) at a defined target region and thus causes small insertions or deletions during non-homologous end joining (NHEJ) repair, which can induce a knockout of a gene of interest. Precise edits or knock-ins can be achieved through homology-directed repair (HDR) of a DSB if a suitable repair template is offered. Due to its high efficiency and ease of use, CRISPR/Cas9 is currently the method of choice for creating genome alterations in animal models. Together with highly developed ARTs, an unlimited set of possible applications arises, making large animals a valuable and very accessible model for gaining a deeper understanding of mammalian preimplantation development ([107], reviewed in [108]).

Genetically modified embryos may be produced by SCNT, where the modifications have been induced in the primary cells that serve as donors of nuclei, or directly in zygotes using zygote injection (ZI) or electroporation. When using SCNT, all embryos have a uniform genotype, show no mosaicism, and donor cells can be screened thoroughly for possible off-target effects, making this the preferred technique for producing genome edited animals (reviewed in [109]). Nevertheless, cloning artefacts (see Section 2.6) that possibly alter developmental mechanisms must be considered and closely monitored by implementing appropriate controls [110]. A high passage number may impair donor cell viability and SCNT success [111,112], which is often the case as cells must be passaged several times in order to produce clonal cells with the desired modifications for SCNT.

A different approach is ZI, where a desired mutation can be induced by injecting the CRISPR/Cas components into a pronucleus or the cytoplasm of a zygote. More recently, successful use of electroporation to manipulate porcine and bovine zygotes has been reported [113,114,115]. Zygote injection or electroporation require less technical effort compared to SCNT and may induce mutations at a high rate. Nevertheless, a tremendous problem is the common effect of mosaicism. When DNA replication precedes CRISPR-mediated genome edition, mosaicism occurs and therefore greatly reduces the odds for generating embryos with a uniform genetic modification. Additionally, the type of mutation is unknown during development and the only narrow genomic material per sample hampers in-depth investigations, especially when further analysis via imaging techniques or transcriptome analyses are needed. Therefore, repeatability and analysis can be a problem when performing ZI [116,117,118,119], but despite the possible drawbacks, it has been recently shown that a knock-in calf can be produced in one step using ZI [120]. Injection of CRISPR/Cas into M-phase oocytes concurrent with ICSI can increase editing efficiency and reduce mosaicism in mouse and human embryos [121].

Genetically modified bovine embryos can be cultured in vitro up to day 8. If later developmental stages are of interest, manipulated embryos may be transferred to the uterus of a recipient cow and flushed non-surgically until shortly before implantation. Van Leeuwen, et al. [122] successfully transferred IVP derived embryos to cows at day 7 and flushed them again at day 11–15, showing the opportunity to examine gene edited embryos at later developmental stages, which at the moment cannot be produced bona fide in vitro.

Other techniques for genetic modification include ICSI- and sperm-mediated or lentiviral gene transfer. Sperm as a vector can be employed during ICSI-mediated gene transfer, where semen is co-incubated with an exogenous transgene before conducting ICSI. This is especially of interest in pig [123,124,125], but the vector may also be used in bovine for IVF [126,127] and even AI for both pig and bovine, as well [128,129]. Unfortunately, these techniques come with high variability in success and unprecise modifications (reviewed in [130]). With lentiviral gene transfer, complex retroviruses are disabled to serve as a vector and can infect both dividing and non-dividing cells. The vector naturally fuses with the cell (oocyte or zygote) and is internalized, making it less damaging compared to microinjection techniques. Lentiviral constructs can be injected into the perivitelline space of a zygote or by co-culture with a zona-free zygote. Transgenesis rates are extremely high with up to 100% in various animal species [131,132]. However, the “cargo size” is limited (6–8 kb at most), multiple integrations at different loci may occur and transgenerational silencing has been reported ([133,134], reviewed in [135]).

## 4. New Insights into Preimplantation Development from Alternative Model Organisms

It is of outmost importance to compare preimplantation development between species to get a comprehensive understanding about different regulatory systems, especially when deciphering the role of various transcription factors during early mammalian development. Two lineage differentiations pave the way during mammalian preimplantation development. First, outer and inner cells of the morula diverge, giving rise to the surrounding CDX2-expressing TE and the ICM. Second, within the ICM the pluripotent NANOG-expressing cells form the EPI and segregate from the differentiated primitive endoderm (PE) or HYPO expressing GATA6 or SOX17 ([136,137,138,139,140,141,142], reviewed in [143]).

In the mouse, the HIPPO/YAP signaling pathway is crucial for the specification of ICM and TE, as outer cells at the 16-cell stage with less cell-to-cell contact polarize and down-regulate the HIPPO signaling pathway. Subsequently, YAP localizes to the nuclei in outside cells and activates TEAD4, leading to the expression of TE-specific genes, such as *Gata3* or *Cdx2* ([144,145], reviewed in [146]). Despite recent gene expression analysis, which indicated differences in early lineage specification in the mouse and other mammals, such as human [147,148,149,150] and cow [151], Gerri, et al. [152] found an evolutionary conserved molecular cascade that initiates TE segregation in human, cow and mouse embryos. HIPPO signaling pathway effectors and TE-associated factors are conserved in cells that initiate the TE program in morula stage embryos of these species, which was confirmed by single-cell RNA-sequencing (scRNA-seq) datasets, immunofluorescence staining and inhibition of modulators of the first lineage segregation. Nevertheless, the group confirmed a different expression pattern of SOX2, a specific marker of the ICM. In the mouse morula, the transcription factor SOX2 is restricted to the inner cells via the HIPPO pathway and considered to be the first marker of pluripotency [153]. In bovine embryos, SOX2 was detected in some blastomeres from the 8-cell stage on, whereas in human embryos SOX2 was expressed in all nuclei. Expression of SOX2 in nuclei of human and cow morulae continues until formation of the expanded blastocyst, where it is finally restricted to cells of the ICM. This is in contrast to mouse, where the restriction starts earlier [152]. 

In the mouse, HIPPO/YAP signaling also plays a crucial role during EPI formation, where the TEAD-YAP dependent variable expression of pluripotency factors, such as SOX2, induces formation of EPI in the ICM. Variations in TEAD activity resulted in a higher proportion of unspecified cells, which are eliminated by cell competition, resulting in a high-quality EPI [154].

The modulation of signaling during lineage differentiations with exogenous factors or inhibitory small compounds is a widely used strategy in developmental studies. During the second lineage differentiation, FGF4/MAPK signaling is vital for PE formation and blocks NANOG expression, resulting in a salt-and-pepper distribution of EPI and PE precursor cells in the ICM (Figure 2). In mouse embryos, inhibition of this pathway leads to a complete ablation of GATA6 and all cells express NANOG [155,156]. However, in bovine embryos inhibition of FGF4/MAPK signaling increases the number of NANOG expressing cells, but only partially blocks GATA6 expression [157]. A more precise marker of the HYPO in bovine embryos is SOX17, as it is mutually exclusive with NANOG already by day 8. Inhibition of MAPK in N2B27 medium showed a dose-dependent response, where increasing the concentration of the inhibitor eventually completely ablated SOX17 expression [158]. When bovine embryos are cultured in the 2i system, which activates the WNT pathway and inhibits MAPK, NANOG expression is increased and GATA6 still present [159].

Therefore, FGF4 signaling is not crucial for GATA6 expression in cattle and a different, so far unknown factor needs to be considered. Interestingly, in human embryos no effect of MAPK inhibition is seen, thus representing an FGF4-independent formation of the HYPO in contrast to other species [157,160]. Similar to cattle, pig embryos treated with MAPK inhibitors showed a severely decreased number of HYPO cells, whereas the number of EPI cells remained unchanged [161,162]. In rabbit, MAPK inhibition has no effect on the expression of EPI markers, but PE markers, such as SOX17, are lost, increasing the proportion of cells that show neither EPI nor PE identity. GATA6 expression on the other hand remained unchanged, indicating that maturation of this cell lineage requires FGF signaling in rabbit ([163], reviewed in [164]). When treating embryos with exogenous FGF4 and heparin, mouse, bovine, pig and rabbit embryos show the same effect: the ICM completely consists of GATA6 or SOX17 expressing cells, suggesting that FGF4 signaling directs GATA6 expression in these embryos ([156,157,162,163], reviewed in [164]). To block the pathway upstream of MAPK, FGF-receptor (FGFR) inhibitors can be used. While in the mouse the ICM again consists only of NANOG expressing cells [165], in human and bovine there is no effect on the lineage precursor cells [157,166]. In the pig, the ICM decreases in cell number while showing an unchanged expression pattern of EPI and HYPO markers [162]. These findings illustrate, that only in the mouse differentiation of the PE is entirely dependent on FGF4/MAPK, while all other examined species seem to regulate this process in an alternative manner. Recently, MAPK/ERK signaling dynamics were investigated more closely via single-cell resolution in the mouse model, which was for the first time able to show a transient inactivation of ERK. First, active ERK was present in both ICM and TE as a consequence of FGF signaling. Subsequently, a subset of mitotic events resulted in short pulses of ERK inactivity in both daughter cells, which later showed elevated NANOG and decreased GATA6 levels. By contrast, non-sister cells exhibited a different signaling pattern, similar to expression patterns reported in embryonic stem cells (ESC) [167,168]. A high ERK activity is found throughout all stages of murine preimplantation development, and only during blastocyst formation a transient ERK inhibition in a subset of cells was found, supporting reports that suggested a low ERK activity resulting in EPI specification, while high ERK activity induces PE formation. This transient ERK inactivation indicates a coordination of cell cycle, signaling and differentiation during embryo formation [168,169]. Another pathway, which plays a vital role in maintaining pluripotency is the Janus kinase/signal transducer and activator of transcription (JAK/STAT) pathway. In the mouse, FGF activates JAK/STAT and increases transcription of ground state pluripotency targets. In bovine, the JAK/STAT pathway was found to be crucial for ICM formation and expression of pluripotency factors, similar to mouse [170].

With regard to OCT4/POU5F1—a key pluripotency transcription factor important for lineage differentiation and maintenance of pluripotency—differences during differentiation between rodent and both human and bovine became apparent. In the mouse, *Oct4* is actively silenced by CDX2 in the TE [151], which is unique, because all other examined species co-express both factors in the TE, reviewed in [108]. These unique regulatory networks might have evolved from different implantation and placentation strategies ([151], reviewed in [171]). OCT4 deficiency in mouse blastocysts causes lack of GATA6 and NANOG persistence [172,173], whereas bovine and human *OCT4*-KO blastocysts lack NANOG, while GATA6 is still expressed [110,150]. In human and mouse, the HYPO or PE specific marker SOX17 is not expressed in the absence of OCT4, indicating a cell-autonomous requirement of OCT4 during the second lineage differentiation [174].

A *SOX2*-KO model in pigs underlined the importance of SOX2 for ICM formation and cell proliferation in porcine early stage embryogenesis in consistence with the mouse model, where targeted embryos formed a blastocoel but failed to form an ICM. Conversely, *Sox2* overexpression in murine 1- and 2-cell embryos led to developmental arrest before the morula stage, whereas in porcine 2-cell embryos, *SOX2* overexpression did not hamper blastocyst formation [175,176,177]. It was speculated that the expression of exogenous *SOX2* via a DNA-lipofectamine system is delayed by ZGA, which starts at 4-cell stage in pigs and thus did not affect early embryonic gene expression [175]. Alternatively, high levels of SOX2 could lead to differentiation as seen in human ESCs, where SOX2 overexpression led to differentiation towards TE cells [178].

With scRNA-seq, another tool is now available to analyze developmental processes in an unprecedented manner. The transcriptome of each single cell within the embryo may now be examined, be it the modulation of various signaling pathways as shown above, or the existence of a naïve pluripotency signature in the morula (day 4–5) and ICM of early blastocyst (day 5–6) in pig [161]. In bovine embryos, scRNA-seq showed an asynchronous blastomere development during the phase of major genome activation [179]. ScRNA-seq opens the way for new approaches to delineate cell fate progression in embryos of large animals. In human and mouse embryos, a characterization of embryogenesis on a genome-wide molecular level has already been reported [180,181]. By comparison of rodent, human, and marmoset embryos, a considerable portion of mouse pluripotency associated factors was not found in the ICM of human and non-human primate blastocysts [180]. As mentioned above, scRNA-seq data was used to declare a conserved TE initiation program in mouse, human, and bovine embryos [152].

Live cell imaging is another new instrument which expands the available toolbox. As bovine and porcine embryos show a lipid-rich dark cytoplasm, time-lapse cinemato-graphy is limited, making confocal microscopy the method of choice when nuclear or chromosomal dynamics are of interest. Yao, et al. [182] performed zygote injections in bovine IVF embryos using mRNA for α-tubulin tagged with enhanced green fluorescent protein (EGFP) as a microtubule marker and histone H2B fused with mCherry as a chromatin marker. This enabled the analysis of nuclear or chromosomal integrity from 1-cell up to blastocyst stage even in spite of the dark cytoplasm. Thus, it was possible to detect a relationship between nuclear abnormalities with embryonic development and morphological quality. A combination of live cell imaging, scDNA-seq and genetic manipulations was used to investigate mitotic divisions and chromosome segregation in bovine embryos, shedding light on the molecular pathways that regulate chromosome fidelity during the error-prone cleavage stage of mammalian embryogenesis [183].

In mouse embryos, live cell imaging revealed new insights in kinetics of transcription factors during cell segregation in vivo. A fluorescence decay after photoactivation assay monitored the location and movement as well as the decay of OCT4, revealing two sub-populations in the early embryo. Cells with slower OCT4 kinetics were more likely to give rise to a pluripotent cell lineage in the ICM, whereas cells with faster OCT4 kinetics segregated to outer cells, indicating that cells of the embryo differ in accessibility of target genes before the physical segregation in inner and outer cells [184]. In the 4-cell embryo, SOX2 engaged in more long-lived interactions with the DNA than OCT4 and varied between cells. Blastomeres displaying more SOX2 binding to DNA were found to contribute more progeny to the pluripotent inner cells of a murine 16-cell stage embryo, thus SOX2-DNA binding predicts cell fate as early as the four-cell stage. This highlights the benefit of this noninvasive imaging method to relate heterogeneities in transcription factor binding with the first cell fate determination ([185,186], reviewed in [187]).

## 5. Conclusions and Outlook

The embryo as research specimen to study preimplantation development in domestic species can be produced in many ways. In choosing the optimal protocol, the focus lies on the production efficiency, reproducibility and the generation of bona fide samples. The embryo that most closely resembles the biology of preimplantation development is produced in vivo without implementation of any ART. While this would generate bona fide samples, the low efficiency especially in uniparous species and no access for experimental procedures make this approach impractical. To increase the yield during in vivo production in cattle, superstimulation offers a long established and easy to perform method. Drawbacks are the animals’ variable response to hormonal treatment and alteration of gene expression patterns in the embryos. IFOT in cows may also increase the yield and provides access to immature COCs before transfer, but variation in blastocyst rates and a high technical effort are disadvantages of this procedure.

IVP of embryos has a great efficiency regarding blastocyst rates and the availability of ovaries from a nearby abattoir is the only requirement for conducting IVP on a regular basis without any animal husbandry. Every step in the development of an embryo can be observed and manipulated during in vitro culture, raising the opportunity to conduct countless different experiments with great sample sizes and thus a high reproducibility. Embryos from IVP show a different transcriptome signature when compared to their ex vivo counterparts and the culture environment has a great impact on development, which must be considered when designing experiments.

In several mammals, including human [188], mouse [189], cow [190,191], pig [192] and sheep [193], it was shown that IVP derived male embryos develop faster to the blastocyst stage compared to female embryos. Variable growth, metabolism, and (epi)genetic programming before implantation may be due to different responses of males and females to changing conditions in environment, including female X-chromosome dosage compensation [194,195]. X-inactivation in mammals is still a topic with open questions and species-specific differences were reported [196,197]. In cloned embryos, abnormal development in both sexes was shown and a connection to variations in X-inactivation was established (reviewed in [97]). These differences between sexes and the effect of in vitro culture on kinetics and epigenetic reprogramming must be considered. Variable mechanisms regarding X-inactivation should be kept in mind when comparing X-linked gene expression of different species.

Components in the culture medium may also bias experiments, e.g., BSA was reported to alter the effect of exogenous FGF4 on mouse embryos [198].

It is possible to perform genetic manipulations directly in the embryo during IVP using zygote injection or electroporation, where mutations can be induced at a high frequency and also more complex alterations can be achieved. Besides using genetically modified animals combined with in vivo development of the embryo, this method offers the specimen closest to the biology of preimplantation development, while enabling genetic studies and the advantages of in vitro culture. Nevertheless, potential drawbacks are the frequent occurrence of mosaicism and only little available material to thoroughly investigate the genotype while simultaneously conducting experiments. SCNT provides the possibility to genetically modify somatic donor cells, which can be clonally expanded and genotyped including possible off-target effects. As every embryo generated then has the exact same genotype, great reproducibility is achieved, and albeit the SCNT procedure being the most artificial technique in producing embryos with its known effects on the embryo, this procedure opens the door especially to more complex experiments. When proper controls are implemented in the experimental setup, the observed effects on embryo development can be traced back to either being due to the actual experiment or the SCNT procedure. A combination of IVP or SCNT with transfer to the oviduct or uterus of a surrogate provides a natural environment and the possibility of studying developmental stages that at the moment cannot be sustained in vitro.

Modern gene editing tools in combination with highly developed ART in domestic large animals offer a platform to challenge the open questions in mammalian preimplantation development. As an example, the role of maternal *Oct4* transcripts stored in the oocyte has been investigated in the mouse using a conditional knockout of *Oct4* in oocytes [172,173]. To achieve this in bovine, female primary transgenic cells expressing Cre recombinase under the control of the *ZP3* promoter, which is active in growing oocytes, and a floxed *OCT4* gene are required. These cells are used for SCNT to produce a cow, from which using OPU a great number of oocytes can be retrieved for IVP of embryos. Together with sperm from a heterozygous *OCT4* knockout bull, it would be then possible to produce embryos where neither maternal nor zygotic OCT4 is present.

Cutting-edge research in the mouse helped us to better understand how the first events of differentiation are induced and regulated and how pluripotency is maintained during preimplantation development. However, species-specific differences during early preimplantation development strengthened the importance of models other than mouse.

Very recently, the first model of a human embryo was introduced, which was developed by reprogramming fibroblasts into in vitro 3D models of the human blastocyst, called iBlastoids, which could help to overcome the scarcity of human material in the future [199].

Nevertheless, bovine and pig models are excellent alternative model organisms to be studied, not only for their similarities to human development, but also for their availability and the established ARTs in combination with the phenomenal tools of gene editing. Together with newly developed analysis techniques, such as single-cell RNA-sequencing or live cell imaging, a comprehensive toolbox is now available which supports the potential of large domestic animals in the field of developmental biology. Bovine and pig embryos are more than an alternative—they are crucial for a complementary understanding of mammalian preimplantation development.

## Figures and Tables

**Figure 1 jdb-09-00015-f001:**
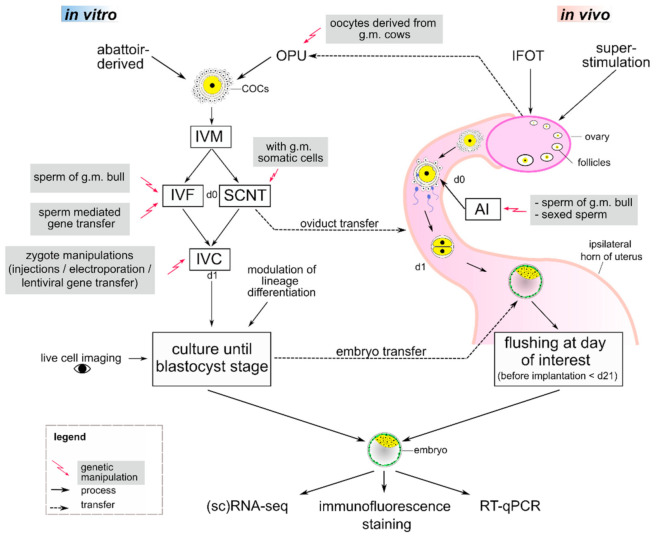
Studying preimplantation development with combined artificial reproduction technologies (ARTs) and genetic manipulation tools in cows. For in vitro production, cumulus-oocyte complexes (COCs) can either be derived from the abattoir or by ovum pick-up (OPU), where oocytes from genetically modified (g.m.) cows may be used. After in vitro maturation (IVM) of COCs, oocytes are in vitro fertilized (IVF) or reconstructed embryos from somatic cell nuclear transfer (SCNT) are activated, marking day 0 (d0) of embryo development. To study preimplantation development, genetic modifications can be introduced during IVF via sperm, by using g.m. donor cells in SCNT, or by direct manipulation of zygotes. These modifications enable, e.g., reverse genetics studies of specific gene functions or tagging of lineage-specific proteins with fluorescent markers. Embryos can then be transferred into the oviduct of a cow or cultured in vitro from day 1 (d1) until day 8. During culture, modulation of lineage differentiation and live cell imaging is feasible. If later stages are of interest, embryo transfer into the uterus can be carried out on day 7. For in vivo production, intrafollicular oocyte transfer (IFOT) or superstimulation increase the oocyte yield and artificial insemination (AI) with g.m. or sexed sperm can be performed. After in vivo development, embryos may be flushed non-surgically from the uterus at the day of interest until shortly before implantation on day 21 (d21). Embryos derived from in vitro or in vivo can be further examined by (single-cell) RNA-sequencing ((sc)RNA-seq), immunofluorescence staining or reverse transcriptase-quantitative PCR (RT-qPCR).

**Figure 2 jdb-09-00015-f002:**
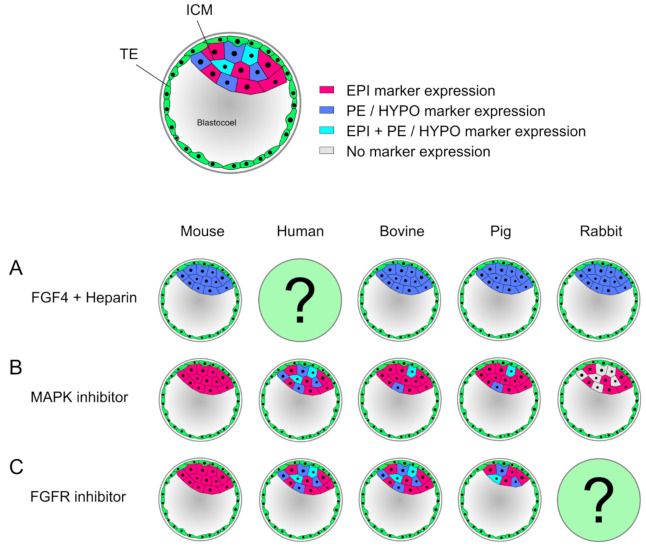
Effects of modulators during second lineage specification vary between mammals. (**A**) Supplementation of FGF4 and heparin leads to a ubiquitous expression of primitive endoderm (PE) and hypoblast (HYPO) markers (dark blue) in mouse and bovine, pig, and rabbit blastocysts, respectively. In human, no experiment has been reported. (**B**) Inhibition of MAPK induces a pan-ICM epiblast (EPI) marker expression (magenta), in sharp contrast to human, where it has no effect. In bovine and pig blastocysts, inhibition of FGF/MAPK pathway does not prevent formation of HYPO precursor cells, though the number of HYPO marker expressing cells is significantly reduced. In bovine, a significant shift towards EPI identity is seen (in pig not significant). Rabbit embryos treated with MAPK inhibitors show no effect on EPI marker expressing cells, but HYPO marker expression was abolished, hence leaving cells with no marker expression (gray), where the identity is unknown. (**C**) Treatment of embryos upstream of MAPK pathway with FGF-receptor inhibitors display a homogenous PE marker expression in mouse blastocysts, similar to MAPK inhibition. In human and bovine embryos, FGFR inhibitors have no effect, whereas in pig embryos, a decreased ICM was reported, but showing a normal distribution of EPI and HYPO markers. In rabbit embryos, the effect is still unknown.

## Data Availability

Not applicable.

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
