# Peer review of "A New Toolbox in Experimental Embryology—Alternative Model Organisms for Studying Preimplantation Development"

_jdb, 2021, doi:10.3390/jdb9020015_

Round 1
Reviewer 1 Report
My comments are written in the attached PDF file.
Please note: 10.1387/ijdb.180414a should be 10.1387/ijdb.180414ap

Author Response
Attached you can find a PDF with our responses.
Thank you!

Reviewer 2 Report
I have now completed reviewing the article “A new toolbox in experimental embryology – alternative model organisms for studying preimplantation development”.
This is well written manuscript describes an overview of different assisted reproductive technologies and their benefits and disadvantages in pig and cattle. There were no major concerns in the manuscript. Only one minor correction needed before the article is suitable for publication.
Because many abbreviations were used in the current manuscript, the reviewer suggests that add an Abbreviation section, and put those abbreviations into the abbreviations section.
Author Response

(The authors gave the same response as above.)

Reviewer 3 Report
General comments
This review is well written, easy to read and understand. It summarizes several decades of development of assisted reproduction techniques in livestock, mainly cattle and pig, where these techniques are the most used or advanced. It offers a very complete overview of the interest of using these particular animal models in addition to the usual mouse mode to study early embryo development. Emphasis is placed at the end of the review on the latest knowledge on the control of differentiation of the first embryonic cell lines by comparing the different models.
This is an excellent review taking into account the most recent studies. More direct comparisons with the human embryo could be done. Moreover future techniques of investigation using microfluidics could be mentioned, but of course choices had to be made as for any review! To my point of view, the only weakness is that it does not mention the early differences in gene expression and metabolism between male and female embryos and the differential impact of embryo culture conditions on those differences.
Specific comments
Abstract
L17 « … and the period until the event of implantation » This might be confusing as implantation occurs much later and at a much advanced embryonic stage in domestic animals by comparison with humans.
L32: idem
State of the ART
L95-98 it might be interesting to precise that, contrary to the situation in human IVF, uterine transfer before the morula/blastocyst stage impairs further embryonic development in cattle and pig. It is why embryo culture is usually performed up to such stages to allow nonsurgical uterine transfer. It is also interesting to note that the first IVF baby is the result of an intra-oviduct transfer of a cleaved embryo. Indeed Bob Edwards was convinced that respecting the physiology was the best way to increase the chance of having a pregnancy!
L114-117 it could be specified that the number of oocytes retrieved per OPU session is highly variable between cows, as is the response to superstimulation treatments. A marker of the population of antral follicles available for both techniques might be the levels of AMH, as it is the case for women.
IVP embryos
L178 I suggest interchanging paragraphs SNCT and IVP (moreover they both are mentioned as 2.5)... because embryo culture for several days is necessary in both techniques and abnormalities such as “large offspring syndrome” have already been observed after IVP in ruminants more than 20 years ago, although they are much more frequent after SCNT. L179-180 layout problem 186-191 oocyte nuclear maturation occurs spontaneously after removal of the COC out of the follicle. Several cocktails can be added to improve cytoplasmic maturation and cumulus expansion, but usually more than 90% nuclear maturation can occur without hormones or growth factors. Serum free media are now used for all IVP steps, which solve several problems encountered with serum, including the high variability between batches.
L195 Percoll separation is not allowed anymore L200 D7 post-insemination or after 6 days of culture
L221-222 the term “zygote” is mainly used for 1-cell stage embryo.
It is important to note that, as in human IVF, the used culture medium can impact the sex ratio of the resulting embryos/calves, or their kinetics of development depending on their sex. For example, serum-containing culture media tended to favor the development of male embryos. This should be added in this paragraph and also mention in the conclusion. A few lines on the X inactivation process which is probably at the origin of those early differences between male and female embryos could be added. The kinetics and mechanisms of X inactivation in female embryos differ between species but might be closer between human and bovine or pig embryos than with mouse embryos (namely concerning the parental imprinting of the process). Today, with the acquired knowledge of the last 10 years, it is no more conceivable not to take into account the sex of the embryo when performing any kind of –omic analysis. Most techniques now allow single embryo and even single cell analysis.
Author Response

(The authors gave the same response as above.)
